



**The mechanism of spray electrification: the waterfall effect**
James K Beattie
School of Chemistry, University of Sydney, Sydney, NSW 2006, Australia
james.beattie@sydney.edu.au
Tel: +61 2 9351 3797
Fax: +61 2 9351 3329



Abstract

13        The waterfall effect describes the separation of charge by splashing at the base of a waterfall.

Smaller drops that have a net negative charge are created, while larger drops and/or the bulk maintain
overall charge neutrality with a net positive charge. Since it was first described by Lenard (1892) the
effect has been confirmed many times, but a molecular explanation has not been available. Application
of our fluctuation-correlation model of hydrophobic hydration accounts for the negative charge
observed at aqueous interfaces with low permittivity materials. The negative surface charge observed in
the waterfall effect is created by the preferential adsorption of hydroxide ions generated from the
autolysis of water. On splashing, shear forces generate small negative drops from the surface, leaving a
positive charge on the remaining large fragment. The waterfall effect is a manifestation of the general
phenomenon of the negative charge at the interface between water and hydrophobic surfaces that is
created by the preferential adsorption of hydroxide ions.






## 1. The waterfall effect


Lenard (1892) reported that the air near the base of a waterfall was negatively charged; subsequent
laboratory experiments indicated that breakup of the water stream is required; a jet of water did not
lead to charge separation (Lenard, 1915). The phenomenon now known as Lenard, waterfall,
balloelectric or spray electrification is well established, having been confirmed at various times and
places (Natanson,1950) (Pierce,1965) (Reiter, 1994) (Laakso, 2007) (Kolarz, 2012). Recent work has
shown that the numbers of both negatively and positively charged small clusters of water molecules less
than 30 nm in diameter are increased but that the negatively charged 'air ions' are at least ten times
more numerous than the positive ones (Kolarz, 2012).

## 2. The search for an explanation


An explanation for the negative charge in the air around the base of a waterfall has been
elusive. Pierce *et al.* wrote "The mechanism producing the space charge remains obscure."(1965).
Laakso and colleagues wrote: "How waterfalls produce ions is far from being completely
understood."(2006). Kamra in 2015 states "The mechanism responsible for production of charge and the
nature of ions produced during splashing of raindrops are not well understood." (Kamra, 2015) Lenard
himself proposed a double layer model in which the dipolar water molecules orient themselves at the
surface of bubbles with the negative end pointing outwards and the positive end pointing inwards. The
latter would attract negative ions which are carried onto small drops when the water breaks up into a
spray. There does not appear to be any independent evidence for this orientation of water dipoles at
the interface (Liu, 2012)(Samson, 2013).

## 3. The hydroxide ion charge







It has been known since 1861 that air bubbles in water are negatively charged and migrate
toward the positive electrode in an electrophoresis experiment (Quincke, 1861). Such measurements
have been refined and repeated many times, always with the same result: the bubbles are negative. Oil
drops in water behave similarly. It has long been known that they spontaneously acquire a negative
charge and migrate toward the positive electrode in a dc electrophoresis cell (Carruthers, 1938). From
the pH dependence of the zeta potential, it was inferred that adsorption of hydroxide ions was
responsible for the negative charge (Carruthers, 1938) (Marinova, 1996). By measuring the pH changes
accompanying the formation of an emulsion with its large surface area, the surface charge density of oils
that have very low solubilities in water is obtained (Beattie, 2004). Its value of ~5 $\mu C\,cm^{-2}$ corresponds
to the adsorption of one hydroxide ion on every 3 $nm^2$ of the oil surface, and is nearly independent of
the identity of the oil. Although the surface charge density at the air/water interface has not been
measured directly, the similarity of the pH dependence of the zeta potentials indicates that it must be
almost the same. Indeed the surface charge density at the Teflon/water interface of 4 $\mu C\,cm^{-2}$
(Preocanin, 2012) indicates that the charge is largely a property of water, almost independent of the
hydrophobic substrate.
3.1 The fluctuation-correlation model
The cause of this preferential adsorption of hydroxide ions remained obscure until 2009 (Gray-
Weale, 2009). We then argued that the hydroxide ion suppresses correlations among fluctuating water
dipoles near the ion and is thus repelled from the bulk by a dispersion force. In a pure polar liquid,
molecules fluctuate in arrangement, and the moments of any two regions of the solvent interact and
become correlated (Figure 1a). This leads to a cohesive force that in part holds the solvent together. The
attraction between two atoms due to correlated, fluctuating, electronic dipoles is the familiar van der
Waals force and has the same mechanism, but in the case of a polar solvent all sources of polarisation



71            A             B

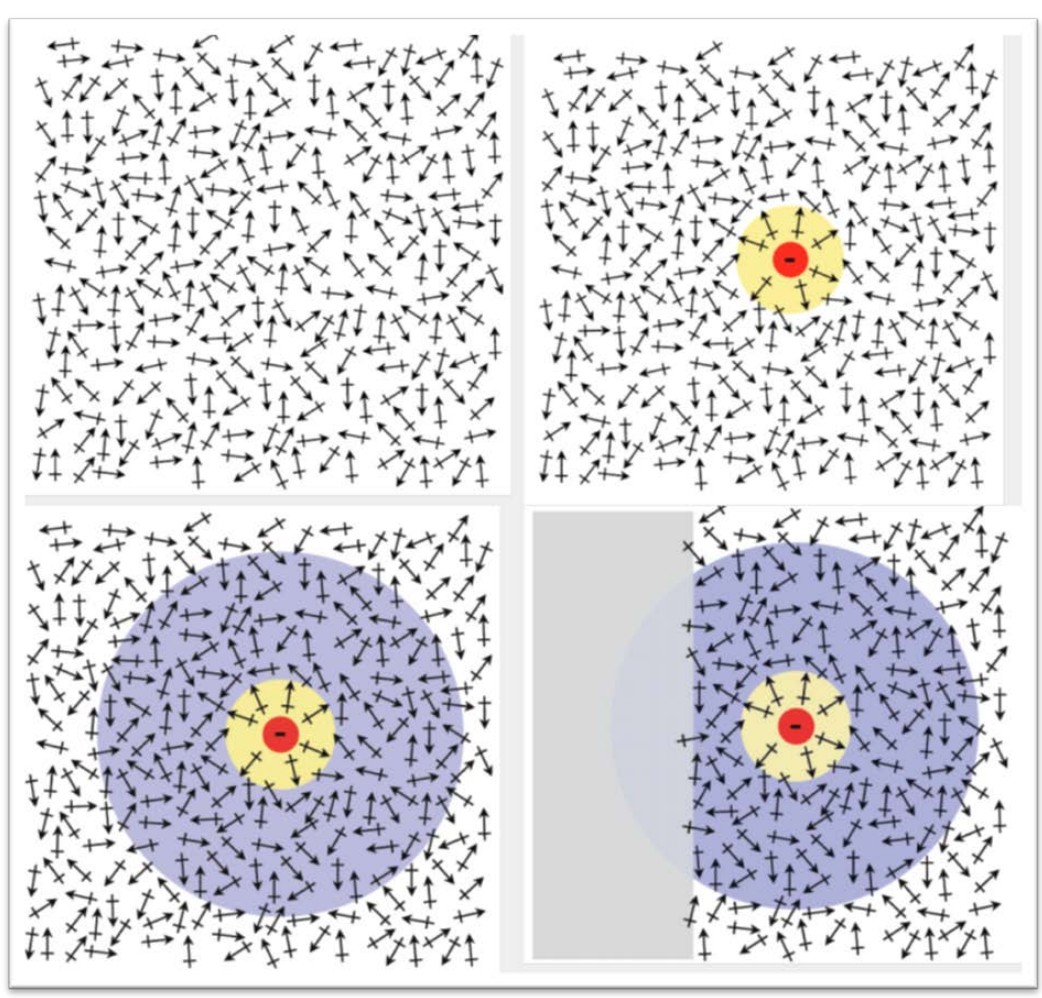


73            C             D

fluctuations, electronic and molecular, are included. An ion in a polar solvent has a solvation shell which
is constrained (Figure 1b) and unable to fluctuate or become correlated with molecules further from the
ion (Figure 1c). This constraint costs free energy because the correlations and consequent attractive
interactions are removed. If the hydroxide ion approaches an interface with a hydrophobe, (a region of



low relative permittivity) the constraint of the solvent molecules near the ion costs less free energy,
because here there are fewer fluctuating solvent molecules within range. This is shown by the overlap of
the fluctuating region with the hydrophobe (Figure 1d). The overlap region contributes to the free-
energy cost of the constrained solvation sphere only when the ion is in the bulk. (Note that the strength
of correlations decays smoothly with distance: the larger circle in Fig 1 is misleading by suggesting a
sharp cut off in correlation range, but it is a useful illustration of the mechanism.) The hydroxide ion is
preferentially attracted to the interface because it has a particularly large dielectric decrement (Gray-
Weale, 2009). An aqueous anion is attracted to an interface with a low-dielectric hydrophobe where
there are fewer water molecules that are excluded from fluctuation correlations with the hydration
waters about the anion.

88        3.2 Molecular explanation

The molecular explanation of the waterfall effect is now completed by recognising that for clean water
with an ionic strength of the order $10^{-5}$ M the double layer thickness is approximately 100 nm. Zilch and
colleagues have given a detailed account of how the rupture of a large drop with an excess of negative
charge at the surface leads to a population of small, negatively charged droplets (Zilch, 2008). They
attribute the accumulation of negative charge to the attraction of hydroxide ions to oriented water
dipoles, but their account can be simply re-expressed in terms of the fluctuation-correlation
explanation. The mechanics of the droplet formation remain the same .
The waterfall effect is hence the consequence of the spontaneous negative charge of the air/water
interface followed by the shear rupture of the overall neutral drop into nanoscopic negative fragments
from the surface with a net positive charge in the core.
**4.    Other manifestations**



The creation of a charged interface by the spontaneous adsorption of hydroxide ions should be
manifest in other phenomena involving water droplets. The interior pH of a millimeter diameter
raindrop will not be affected by the adsorption of hydroxide ions to the surface because the surface to
volume ratio is too small. But smaller fog and mist drops with larger surface-to-volume ratios could have
a more acidic interior. The increased acidity of small water drops has been observed by single-molecule-
sensitive fluorescence resonance energy transfer of labelled rna in freely-diffusing droplets . The spectra
of the rna in 230 nm drops shifts to that characteristic of weakly acidic solutions of pH 4 from that
observed in neutral solution without droplets.(Rahmenseresht, 2015)
The charge on the air/water interface has a profound effect on the rates of reactions at the
surface. In a series of elegant experiments with a water jet, Colussi and colleagues (Mishra, 2012) have
demonstrated that the isoelectric point for acid-base reactions between the water surface and gaseous
reactants is shifted from pH7 to pH3, just as is found for acid-base equilibria at the air/water and
oil/water interfaces. The agreement between these very different measurements provides strong
support for the hydroxide explanation and eliminates some alternative interpretations.

### 5. Conclusion

In summary, recognition of the role of the hydroxide ion in the spontaneous charging of the
air/water interface provides the molecular explanation for the waterfall effect that has been missing
since the 19[th] century. Together with the recent observations of charged nanodrops by Laakso and
Tammet and their co-workers, and the description of the fragmentation of neutral into charged drops,
the explanation of this effect is complete.

### Acknowledgements





This work was supported by the Australian Research Council. The author thanks Dr. Richard O'Brien for
many helpful discussions.

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
