# Peer review of "The mechanism of spray electrification: the waterfall effect"

_Atmospheric Chemistry and Physics, 2015_

## Referee Comment (RC1) · Anonymous Referee #3 · 5 Jul 2016

Referee report on "The mechanism of spray electrification: the waterfall effect" by J. K. Beattie

General recommendation

The author investigated mechanism of water spray electrification, so-called balloelectric effect or Lenard effect. It is generally known that the breaking or splashing of water drops, as well as the bursting of bubbles in a water-air interface can generate large amounts of negatively charged particles and, therefore, a negative space charge in the atmosphere during rainfall and close to e.g. natural waters, the sea waves and waterfalls (Adkins, 1959; Gathman and Hoppel, 1970; Levin, 1971; Reiter, 1994). The Lenard effect (Lenard, 1892) occurs when drops of pure water break up into numerous fine negative droplets. The Lenard effect has been observed in several location when

measuring small and intermediate ambient ions (Tammet et al., 2009, Laakso et al., 2007; Kolarz et al., 2012; Kamra et al. 2015) which I can personally confirm.

What makes this study to stand out is the attempt to actually explain theoretically the Lenard effect on a molecular level. Author offers an alternative explanation to Lenard's original explanation. These small negative ions may have a role in atmospheric particle formation in small scale. The initial formation processes, i.e. nucleation, which occur in the particle diameter range well below 10 nm are the subjects of extensive research and some of these processes favor the negative charge. Usually, the atmospheric newly formed charged particles and the bursts of small and intermediate ions, which are generated during the rain, on the sea waves or at the base of a waterfall, are considered as separate phenomena. On the other hand, the recent airborne studies have shown that small charged particles are observed as an outflow from the clouds. Interesting question is whether those small ions are nucleated or formed due to by the splashing of water (water droplet breaking in air).

There are some issues to be resolved, mainly related to some corrections and providing more details, which help understanding the methods and the results. How well does the experiments support existence of these negative ions generated from splashing water? In the light of the field and laboratory observations, is the theoretical size distribution of these charge carriers in agreement with the measured/observed size distribution? What is still unclear to me is the large drops and/ or the bulk with positive net charge?

Specific comments

Page 3, line 29-34: To add recent studies by Tammet et al. (2006) and Kamra et al. (2015 and references therein) on the direct observations of small balloelectric ions.

Page 3, line 32-22: In atmospheric science, the small clusters usually refer to air ions smaller than 3 nm and larger ions are called intermediate ions (3-7 nm) and large ion or charged particles (larger than 7 nm). Therefore, ". . .clusters of water molecules less than 30 nm in diameter. . ." should be replaced with "clusters of water molecules and

water droplets less than 30 nm in diameter".

Page 6, lines 89-95: What is the actual size distribution of these charge carriers? Does it agree with the atmospheric observations? E.g. Tammet et al. (2007) observed balloelectric ions in at ∼7 nm whereas Kolarz et al. (2012) observed that the most abundant sizes were nanometer sized ions at 2 nm and submicrometer ions at 120 nm.

Page 6, lines 96-98: "The waterfall effect is hence the consequence of the spontaneous negative charge of the air/water interface followed by the shear rupture of the overall neutral drop into nanoscopic negative fragments from the surface with a net positive charge in the core." Please illustrate this as well in a figure (e.g. Fig. 2 or add to Fig 1). The figure would be a great help to show what is the negative "surface" and positive "core" of the droplet.

Page 7, lines 115-119: Laakso and Tammet are not the only ones studying this effect. Therefore, consider modifying the sentence.

Page 7, lines 115-119: In addition, could you comment how is this theory applied? Can it also be used to explain other bursts of intermediate ions generated by the splashing of water (e.g. generated by rainfall and sea-spray) so called the balloelectric ions?

Technical correction

Type check manuscript, e.g. references in the text, and modify the manuscript to match the ACP guidelines: www.atmospheric-chemistry-and-physics.net/for_authors/manuscript_preparation.html

Page 8, line 125: Reference list is missing following citations which are mentioned in the text: Tammet et al. 2009 and Kamra et al. 2015.

Page 5, line 71: Fig. 1 is missing caption, or? The figure was improved well from the previous version. The caption should explain the steps introduced in the figure.

References missing:

Kamra, A. K., A. S. Gautam, and D. Siingh: Charged nanoparticles produced by splashing of raindrops, J. Geophys. Res. Atmos., 120, 6669–6681, doi:10.1002/2015JD023320, 2015.

Tammet, H., Hõrrak, U., and Kulmala, M.: Negatively charged nanoparticles produced by splashing of water, Atmos. Chem. Phys., 9, 357-367, 2009.
* * *

---

## Referee Comment (RC2) · Anonymous Referee #4 · 18 Aug 2016

While this manuscript deals with a potentially interesting research topic, it lacks several features that are required for a scientific research article. A scientific research articles should contain the following elements: introduction, description of used methods, representation of new results, discussion of these results, and conclusions. This paper has several problems associated with this:

First, this paper lacks entirely the methods section that should provide information on the sources of data used in this paper.

Second, partly because of the first issue, it remains unclear where the main results (figure 1 and associated discussion) come from, and are these results even produced in this particular study.

Third, I have some difficulty in separating when authors represent new results, when

they discuss earlier results, and when they discuss the result obtained in this study.

Fourth, sections 1 and 3 and part of section 3 contain introductory material that is commonly under the title "Introduction" in a scientific article.

Fifth, the "conclusions" section is not real conclusions: the first sentence is introductory material, the second one mainly discussion type material.

Finally, the prior research is not acknowledged in a proper way in the paper. Citations are missing from many places where a citations should exist, and many of the existing citations are imperfect (like writing Tammet and co-workers without specifying the actual scientific articles).

As a result of the problems mentioned above, I cannot recommend accepting this paper for publication in ACP.

―――――――――――――――――――――

---

## Author Comment (AC1) · 20 Sep 2016

REFEREE 3 recognises that this paper does not conform to the style used to report new experimental results, but instead describes a new interpretation of observations that began with Lenart in the 19th century and include those reported just now in 2016. The paper argues that the waterfall effect, the presence of clusters of water molecules and droplets less than 30 nm in diameter with negative charges consequent on the splashing at a waterfall, is one specific example of the general formation of a double layer with a negative inner charge that occurs spontaneously when water forms an interface with a low dielectric hydrophobe. The thickness of the double layer in water at the natural pH of 5.6 is of the order 200 nm. Hence fragments of a larger drop formed by shear of the surface will have a net negative charge. Charge neutrality requires that the remainder of the larger drop and the bulk water formed by these must be positively

charged. Ultimately condensation of the smaller negative drops with the larger positive drops or pools restores the system to its original neutral state.

Zilch and colleagues have given a detailed account of how the rupture of a large drop with an excess of negative charge at the surface leads to a population of small, negatively charged droplets (Zilch et al., 2008). They attribute the accumulation of negative charge to the attraction of hydroxide ions to oriented water dipoles, but their account can be simply re-expressed in terms of the fluctuation-correlation explanation. The mechanics of the droplet formation remain the same.

The author is not aware of a theoretical model that predicts the size distribution of the negative ions. The distribution is likely to be very sensitive to the conditions in which it is formed, being a dynamic process. For example, a large difference is expected between the size distribution found in fresh water and that in sea water measured with identical parameters, because the properties of the water are significantly different, in pH, viscosity, ionic strength, etc. In response to specific comments of Referee 3:

References to Tammet et al and to Kamira et al added.

The names describing the different sized charges species has been adopted.

As discussed above, the size distributions under different conditions are expected to differ.

The added Figure from Zilch contributes to clarification of the process.

The last sentence referring to Laakso and Tammet has been replaced by several commenting on the generality of the concept presented in the paper.

  Referee 4 is disturbed by the departure of the paper from the conventional format used to describe new experimental results. As discussed above, this paper examines to consequences of a new idea, the origin and application of the spontaneous adsorption of hydroxide ions to nanoscopic droplets formed in a waterfall and by other means. The only 'new result' is the description of the effect of the adsorbed hydroxide

[Figure]

[Figure]

**Fig. 1.**

---

## Author Comment (AC2) · 20 Sep 2016

Referee 4 is disturbed by the departure of the paper from the conventional format used to describe new experimental results. As discussed above, this paper examines to consequences of a new idea, the origin and application of the spontaneous adsorption of hydroxide ions to nanoscopic droplets formed in a waterfall and by other means. The only 'new result' is the description of the effect of the adsorbed hydroxide. The paper could, of course, at the expense of a much increased length, describe all of the observations reported in the literature since the original and perceptive one by Lenart in 1892. But this is unnecessary, especially so in this case, because there is consensus on the matter – the droplets are negatively charged. There are no new experimental results or data. The only new 'result' is the new interpretation of the origin of the charge; in this sense the entire paper is the Result and associated Discussion.